# Prognostic Impact of Total Lesion Glycolysis (TLG) from Preoperative ^18^F-FDG PET/CT in Stage II/III Colorectal Adenocarcinoma: Extending the Value of PET/CT for Resectable Disease

**DOI:** 10.3390/cancers14030582

**Published:** 2022-01-24

**Authors:** Sea-Won Lee, Hye Lim Park, Nara Yoon, Ji Hoon Kim, Jin Kyoung Oh, Jae Ho Buyn, Eun Kyoung Choi, Ji Hyung Hong

**Affiliations:** 1Department of Radiation Oncology, Eunpyeong St. Mary’s Hospital, College of Medicine, The Catholic University of Korea, Seoul 03312, Korea; lords_seawon@hotmail.com; 2Division of Nuclear Medicine, Department of Radiology, Eunpyeong St. Mary’s Hospital, College of Medicine, The Catholic University of Korea, Seoul 03312, Korea; prhlim@gmail.com; 3Department of Pathology, Incheon St. Mary’s Hospital, College of Medicine, The Catholic University of Korea, Incheon 21431, Korea; waxggul@gmail.com; 4Department of General Surgery, Incheon St. Mary’s Hospital, College of Medicine, The Catholic University of Korea, Incheon 21431, Korea; samryong@catholic.ac.kr; 5Department of Radiology, Incheon St. Mary’s Hospital, College of Medicine, The Catholic University of Korea, Incheon 21431, Korea; jinkyoung.oh@gmail.com; 6Department of Internal Medicine, Incheon St. Mary’s Hospital, College of Medicine, The Catholic University of Korea, Incheon 21431, Korea; jhbyun37@catholic.ac.kr; 7Department of Internal Medicine, Eunpyeong St. Mary’s Hospital, College of Medicine, The Catholic University of Korea, Seoul 03312, Korea

**Keywords:** colorectal cancer, PET/CT, total lesion glycolysis

## Abstract

**Simple Summary:**

PET/CT is rarely performed initially in resectable colorectal cancer and is usually considered for detection of distant metastasis. However, we perceived another potential role of PET/CT in addition to diagnosis and staging, which is providing prognostication of the oncologic outcome by PET parameters extracted from initial PET/CT before surgery. This study evaluated the prognostic role of preoperative 18F-FDG PET/CT in 327 stage II/III colorectal cancer patients and comprehensively investigated the PET parameters with multiple threshold levels to select optimal parameters most robustly related to DFS. Several PET parameters including SUVmax, MTV2.5, MTV3, TLG2.5, TLG3, and TLG30% were significantly related to DFS, with TLG2.5 retaining statistical significance in multivariate analysis with other clinicopathologic prognostic factors. Prognostication with PET/CT at the time of initial diagnosis has substantial benefits over pathologic prognostic factors available only after surgery by giving oncologists an opportunity to consider treatment intensification or de-intensification before initiation of treatment.

**Abstract:**

We investigated the prognostic role of metabolic parameters from preoperative ^18^F-FDG PET/CT in stage II/III colorectal adenocarcinoma. A total of 327 stage II/III colorectal adenocarcinoma patients who underwent curative resection were included. The maximal standard uptake value (SUVmax), metabolic tumor volume (MTV), and total lesion glycolysis (TLG) were analyzed for optimal cut-offs and their effect on DFS. Differences in DFS rates and hazard ratios for DFS between cut-offs were statistically significant in SUVmax, MTV2.5, MTV3, TLG 2.5, TLG3, and TLG30%. Factors significantly related to DFS in univariate Cox regression were age, sex, stage, preoperative CEA, SUVmax, MTV2.5, MTV3, TLG2.5, TLG3, and TLG30%. Age, sex, preoperative CEA, and TLG2.5 (*p* = 0.009) sustained statistically significant difference in multivariate analysis. The 1-, 3-, and 5-year DFS rates for TLG2.5 ≤ 448.5 were 98.1%, 79.6%, and 74.8%, significantly higher than 78.4%, 68.5%, and 61.1% of TLG2.5 > 448.5, respectively (*p* = 0.012). TLG, a parameter indicating both the metabolic activity and metabolic volume, was the strongest predictor independently associated with DFS, among several PET parameters with statistical significance. These results suggest the potential prognostic value of preoperative ^18^F-FDG PET/CT in stage II/III resectable colorectal cancer.

## 1. Introduction

Colorectal cancer (CRC) is the second most common cause of cancer death in the United States [1]. Despite a decrement in incidence due to wide distribution of colonoscopy, it is still a major source of morbidity and mortality by malignancy. While colonoscopy remains the gold standard for diagnosis of CRC by allowing pathologic confirmation in addition to morphologic assessment, computed tomography (CT) or magnetic resonance imaging (MRI) have been the main imaging modalities for initial staging. On the other hand, 2-deoxy-2-fluoro-D-glucose (^18^F-FDG) positron emission tomography/computed tomography (PET/CT) is rather considered under restricted circumstances such as suspicion of synchronous distant metastasis. Another reason it is rarely suggested for initial staging is the limited accuracy due to FDG uptake by non-cancerous lesions or physiologic processes typical of the bowel [2]. The low incidence of distant metastasis at initial diagnosis of CRC and the low sensitivity of ^18^F-FDG PET/CT for nodal metastases are a few more reasons that add modest value to PET/CT for initial staging, which is why it is not routinely indicated especially for resectable CRC [3,4].

Stage 0/I CRC has very low risk of recurrence, while stage IV bears poor prognosis after the diagnosis. In contrast with these early or late stages with fairly predictable outcome, stage II/III CRC after resection are reported to have recurrence rates ranging from approximately 20% to 40% [5,6], which calls for an effective tool to predict prognosis. However, most of the reported prognostic factors such as histologic grade, lymphovascular invasion, neural invasion, number of dissected lymph nodes, or microsatellite instability (MSI) caused by defects in DNA mismatch repair genes including MSH-2 and MLH-1 can be obtained with certainty from the final pathologic specimen only after surgery [7,8,9]. Alternatively, prediction of prognosis before initiation of treatment may offer substantial benefits. Consideration of treatment intensification for poor prognostic patients with metabolically aggressive tumor identified using preoperative ^18^F-FDG PET/CT may serve as an opportunity to improve disease outcome by adhering to the newer paradigm of total neoadjuvant therapy. Discussing options for intensive treatment beforehand can help patients better understand the entire course of treatment and augment therapeutic compliance.

Previous studies on the prognostic value of preoperative ^18^F-FDG PET/CT in colorectal cancer report various results. The controversy may be attributable to several factors, including different parameters among studies. PET parameters may be categorized into: (1) the parameters indicating metabolic activity such as maximal standard uptake value (SUVmax); (2) the parameters indicating metabolic volume such as metabolic tumor volume (MTV); and (3) the parameters indicating both the metabolic activity and metabolic volume such as total lesion glycolysis (TLG). These PET parameters are obtained by using certain thresholds such as SUV ≥ 2.5 or ≥ 3.0 or ≥ 30% or ≥ 40% of the SUVmax value within the margin of the contoured tumor. In addition to heterogeneous patient cohorts and different treatments, the thresholds and the cut-off levels for PET parameters adopted in previous studies are variable and thus may have generated controversial results. In order to overcome these limitations, we evaluated the prognostic role of preoperative ^18^F-FDG PET/CT in a relatively homogeneous patient cohort under uniform treatment and comprehensively investigated the PET parameters for metabolic activity, metabolic volume, and both features measured with different threshold levels to select optimal parameters most robustly related to disease-free survival. This study was incubated with anticipation of broadening the perhaps currently underestimated value of ^18^F-FDG PET/CT at initial diagnosis of stage II/III resectable CRC.

## 2. Materials and Methods

### 2.1. Patients

A total of 486 consecutive colorectal cancer patients underwent primary surgical resection at a tertiary university hospital from February 2009 to December 2013. The inclusion criteria were as follows: (1) pathologically confirmed as adenocarcinoma at colon or rectum by full colonoscopy; (2) stage II/III according to the 8th edition of the American Joint Committee on Cancer TNM staging system [10]; (3) acquisition of preoperative ^18^F-FDG PET/CT scan; and (4) no uncontrolled infection before surgery. The exclusion criteria were as follows: (1) histology other than adenocarcinoma such as neuroendocrine tumor, lymphoma, and squamous cell carcinoma; (2) coexistence or metachronous development of other primary cancer; and (3) underlying familial adenomatous polyposis or hereditary nonpolyposis colorectal cancer. Tumors located in cecum, ascending colon, and proximal two-thirds of the transverse colon were categorized as right colon cancer and tumors from the distal third of transverse colon to sigmoid colon were categorized as left colon cancer by CT finding. Approval for this study was obtained from the Institutional Review Board (No. OC16RISI0136). Informed consent was waived due to the retrospective design. This study was designed and performed in compliance with the Reporting Recommendations for Tumor Marker Prognostic Studies (REMARK) criteria [11].

### 2.2. ^18^F-FDG PET/CT Acquisition

All PET/CT scans were performed on a dedicated PET/CT scanner (Discovery STe, General Electric Healthcare, Milwaukee, WI, USA). All patients fasted for at least 6 h, and blood glucose levels were less than 140 mg/dL before intravenous administration of ^18^F-FDG. A dose of approximately 5.5 MBq/Kg of ^18^F-FDG was intravenously administered. PET images were acquired from the cerebellum to the proximal thighs in 3-D mode 60 min after injection of FDG immediately after acquiring a CT scan. PET images were reconstructed by an iterative reconstruction algorithm using the CT images for attenuation correction.

### 2.3. ^18^F-FDG PET/CT Analysis

All PET/CT images were transferred to the GE Xeleris workstation, which produced multiplanar reformatted images and displayed attenuation-corrected PET images, CT images, and PET/CT fusion images. Two experienced nuclear medicine physicians (E.K.C. and J.K.O.) who were blinded to the patients’ identity and clinical outcome reviewed the PET/CT images with visual assessment and semi-quantitative methods. If there was any disagreement in the visual analysis between the physicians, it was resolved by consensus. In vivo assessment of glucose metabolism was estimated using three metabolic parameters: SUVmax, MTV, and TLG.

SUVmax was measured from PET images by placing a spherical a volume of interest (VOI) at the site of primary tumor. The MTV values were calculated using fixed SUV thresholds of 2.5 and 3.0 and relative thresholds of 30% and 40% of SUVmax. TLG was calculated as the volume of MTV multiplied by the average SUV of the MTV.

### 2.4. Diagnostic Work-Up and Treatment

Physical examination and blood tests including complete blood count, blood chemistry, and carcinoembryonic antigen (CEA), as well as chest and abdominopelvic computed tomography (CT) were performed preoperatively together with colonoscopy and ^18^F-FDG PET/CT. For rectal cancer, digital rectal examination and rectal magnetic resonance imaging (MRI) were included. Surgical resection of primary tumor and mesenteric, mesorectal, and/or lateral pelvic lymph node dissection were performed by board-certified, experienced colorectal surgeons. In addition to curative surgical resection, chemotherapy (and radiotherapy in case of rectal cancer) before or after surgery was allowed, in accordance with the current treatment guidelines under collaboration of a multidisciplinary team for colorectal cancer comprised of surgical oncologists, medical oncologists, radiation oncologists, pathologists, diagnostic radiologists, and nuclear medicine specialists.

### 2.5. Analysis of Clinicopathologic Factors

Patients’ clinicopathologic features were collected retrospectively from medical records. The clinicopathologic factors known to be related to prognosis of colorectal cancer such as T stage, N stage, differentiation, lymphatic invasion, vascular invasion, neural invasion, resection margin, KRAS and BRAF mutation, and expression of MSH2 and MLH1 for determination of microsatellite instability (MSI) were assessed by board-certified colorectal specialized pathologists [7,8]. Formalin-fixed and paraffin-embedded tissue blocks were sectioned in order to extract genomic DNA. Pertinent mutations for KRAS and BRAF genes were identified with cycle sequencing. Relevant primers were used accordingly for gene amplification with real-time polymerase chain reaction (PCR), in accordance with the guidelines for gene testing [12,13,14,15]. Immunohistochemistry of MSH2 and MLH1 were performed by staining slides made from tissue sections representative of tumors using mouse monoclonal antibodies [16]. Loss of expression was reported when the nuclear staining of tumor cells was absent to the level of positively labeled non-neoplastic cells [17]. MSI status was defined as high if at least one of MSH2 or MLH1 had loss of expression. MSI-low state was defined as positive expression of both MSH2 and MLH1 [17].

### 2.6. Follow-Up

Patients were followed up at 3- to 6-month intervals for the first 2–3 years, every 6- to 12- months for the next 2–3 years, and then yearly thereafter. Evaluation during follow-up consisted of physical examination during clinical interview, colonoscopy, serum tests including CEA, and imaging with CT and/or MRI at appropriate intervals. Biopsy for histologic confirmation of recurrence was performed at the physician’s discretion.

### 2.7. Statistical Analyses

The continuous values of PET parameters including SUVmax, MTV, and TLG are described as mean ± standard deviation (SD) (range). Survival was calculated from the date of surgical resection. Disease-free survival (DFS) was defined as the time to detection of first recurrence or death. Overall survival (OS) was defined as the time to death from any cause. Survival rates were estimated with Kaplan–Meier analysis. The cut-off level for each PET parameter, which most significantly segregates DFS, was determined with the maximally selected chi-square test (‘maxstat’ package) by R software, version 4.1.1 (R for Statistics Computing, Vienna, Austria). Cox proportional hazards regression analysis was performed to obtain the hazard ratio (HR) and 95% confidence interval (CI) of prognostic factors for DFS. Multivariate analysis was executed with factors that were statistically significant in univariate analysis in order to test for independent association with DFS. Statistical significance was defined at *p* < 0.05, and statistical analyses were performed using the IBM SPSS (Statistical Package for Social Sciences) software for Windows, version 24.0 (IBM Corp., Armonk, NY, USA).

## 3. Results

### 3.1. Patient, Tumor, and Treatment Characteristics

Among 486 colorectal patients who underwent primary surgical resection, 327 patients diagnosed with resectable stage II/III colorectal cancer with preoperative FDG PET/CT satisfied the inclusion criteria and were analyzed in this study (Table 1). Males were more predominant, and majority of the diseases were left-sided. Although over half of the tumors were small (≤5 cm), 10% presented with obstruction. A majority of patients had ≥13 lymph nodes removed at surgery. Most of the tumors were moderately differentiated. Clear resection margin was achieved in majority of patients. While the majority of patients were tested for MSI, less than half of the patients were tested for KRAS or BRAF mutation. Approximately half of rectal cancer patients underwent radiotherapy and over three-quarters of patients received chemotherapy.

### 3.2. Survival Outcome

The patients were followed up for a median of 45.8 months (range: 0.4–81.9). Median OS was not reached, and 32 patients had expired at the time of analysis. The 1-, 3-, and 5-year OS rates were 97.5%, 93.7%, and 87.2%, respectively. Median DFS was 43 months (range: 0–82). At the time of analysis, 68 patients had experienced recurrence (24 locoregional and 46 distant), among which 2 patients had concomitant local and distant recurrence. The most common site of distant metastasis was lung (*n* = 20), followed by liver (*n* = 13). The 1-, 3-, and 5-year DFS rates were 89%, 77.7%, and 74.6%, respectively. 

### 3.3. Effect of PET Parameters on DFS

The value of PET parameters that most significantly discriminated DFS were determined as the cut-off levels (Table 2). The differences in DFS rates between cut-off levels were statistically significant in SUVmax, MTV2.5, MTV3, TLG2.5, TLG3, and TLG30%, and at least marginally significant in rest of the PET parameters (Table 2). The hazards ratios for DFS between cut-off levels were also significantly and marginally different in the same parameters as above (Table 2). The mean ± SD values of SUVmax, MTV2.5, MTV3, TLG2.5, TLG3, and TLG30% were 7.57 ± 4.41, 49.14 ± 46.34, 40.10 ± 39.40, 316.95 ± 385.38, 290.28 ± 366.67, and 241.24 ± 278.27, respectively.

### 3.4. Factors Associated with DFS

The PET parameters were significantly associated with DFS in univariate analyses; SUVmax, MTV2.5, MTV3, TLG2.5, TLG3, and TLG30% were analyzed with other clinicopathologic factors for association with DFS (Table 3). The factors significantly related to DFS in univariate Cox regression analysis were age, sex, stage, preoperative CEA level, SUVmax, MTV2.5, MTV3, TLG2.5, TLG3, and TLG30% (Table 3). Older patients were more likely to recur than younger patients. Female patients were less likely to experience recurrence than male. Stage III was more than twice as likely to recur compared with stage II. The results were similar for patients with higher preoperative CEA level in comparison with patients with lower CEA level. Patients with higher SUVmax had over 60% greater risk of recurrence compared with patients with lower SUVmax. Patients with higher MTV2.5 had nearly twice the risk of recurrence than the lower counterpart. Patients with higher MTV3 had over twice the risk of relapse compared with the lower counterpart. The results were similar in patients with higher TLG2.5 and TLG3. Patients with higher TLG30% also had over three-quarters the risk of recurrence than the lower counterpart. Patients with positive margin and number of removed lymph nodes less than 13 had higher risk of recurrence, but the difference was marginal.

Among the factors significantly associated with DFS in univariate analysis, age, sex, preoperative CEA level, and TLG2.5 sustained a statistically significant difference in multivariate analysis (Table 2). Older patients had over 60% greater risk of recurrence than younger patients. Female patients had 55% of the risk for recurrence compared with males. The results for preoperative CEA level were similar with univariate analysis, and patients with higher CEA level had more than twice the risk of recurrence than the lower counterpart. Patients with higher TLG2.5 were nearly twice as likely to recur compared with patients with lower TLG2.5, with strong statistical significance. The 1-, 3-, and 5-year DFS rates for patients with lower TLG2.5 were 98.1%, 79.6%, and 74.8%, significantly higher than 78.4%, 68.5%, and 61.1% of patients with higher TLG2.5, respectively (Figure 1).

## 4. Discussion

This study demonstrates the potential prognostic impact of preoperative ^18^F-FDG PET/CT by selecting optimal PET parameters using several thresholds that best predict DFS in stage II/III resectable colorectal adenocarcinoma. The prognostic value of PET parameters obtained from diagnostic ^18^F-FDG PET/CT for predicting recurrence of malignancies such as lymphoma, lung, head and neck, and cervical cancers have been actively studied [18,19,20,21,22]. However, research regarding the prognostic significance of PET parameters in the field of colorectal cancer are less extensive. Owing to the underlying physiologic FDG uptake by bowel mucosa, the malignant potential of incidental FDG-avidity and its discrimination from inflammatory bowel disease or accuracy of staging with PET/CT are much more common in the literature of colorectum.

Most of the previous studies on the prognostic significance of PET parameters in colorectal cancer adopt surrogate endpoints such as pathologic tumor response rather than survival outcome itself as the endpoint, which explains the relative abundance of such studies for rectal cancer undergoing neoadjuvant chemoradiotherapy compared with colon cancer [23]. The studies that examined the effect of PET parameters on survival outcome such as DFS or OS usually derived significant PET parameters using PET/CT obtained after neoadjuvant treatment rather than pretreatment PET/CT [24]. In addition, previous studies on the prognostic significance of PET parameters from the pretreatment PET/CT report discordant results (Table 4). Some studies report that PET parameters had no significant association with survival outcome [23,25,26,27,28]. Among those that did report significant association between PET parameters and survival outcome, the significant parameters were variable, including SUVmax, MTV, TLG, or TLR, measured from different regions of interest such as primary tumor, regional nodes, or even bone marrow [29,30,31,32,33,34,35,36,37,38,39,40,41]. Although the majority of the studies used the receiver operating characteristic (ROC) curve to calculate cut-off levels, other methods such as mean, median, or maximal chi-square analysis were used as well. The survival outcome selected as endpoints was also different between studies; DFS or OS rates at different follow-up periods and mean or median survival time in months were used. We considered the major cause of this variability in outcome of previous reports to be due to different clinical setting as a result of different inclusion criteria between studies, especially the stage. Many previous studies on resectable disease still included a fraction of stage IV patients; thus, they analyzed patients receiving a variety of primary treatments. Therefore, we studied the patients under a specific clinical setting of stage II/III colorectal cancer and thoroughly investigated the effect of different kinds of PET parameters with multiple thresholds on DFS in attempt to further clarify the above-mentioned variability.

According to our data, the majority of the tested PET parameters with different thresholds including SUVmax, MTV2.5, MTV3, TLG2.5, TLG3, and TLG30% discriminated DFS with statistical significance and with marginal significance for MTV30% and TLG40%. The significant correlation of numerous PET parameters with DFS irrespective of parameter type (indicating metabolic activity and/or metabolic volume) and threshold levels for parametric measurement underscores the potential prognostic value of PET parameters obtained from preoperative ^18^F-FDG PET/CT. When the PET parameters significantly related to DFS in univariate analysis were tested for independent association with DFS in multivariate analysis, including other clinicopathologic factors, TLG2.5 was the only significantly associated PET parameter (*p* = 0.009). The prognostic factor with most powerful association was CEA level (*p* = 0.001), which is an already well-established prognostic factor. This result again demonstrates the validity of our data by corroborating previous observations. Our data sufficiently suggest the additional value of PET parameters for prognostication. SUVmax is a parameter for metabolic activity, and MTV is a volumetric parameter, while TLG2.5 is a parameter indicating both the metabolic activity and metabolic volume—which may explain the maintenance of statistical significance only for TLG2.5 (cut-off: 448.5) at multivariate analysis. Although TLG2.5, TLG3, and TLG30% were significantly related to DFS in univariate analysis, and TLG2.5 was the only TLG which retained significance in multivariate analysis, which threshold within TLG best demonstrates the prognosis remains to be seen in future studies. For locoregionally advanced, resectable colorectal cancer, the PET parameter demonstrating both the metabolic activity and metabolic volume such as TLG significantly prognosticated DFS.

Our results are in line with previous studies by demonstrating the significant association with DFS of different types of PET parameters, including SUVmax, MTV, and TLG, which have been previously reported to be related to survival outcomes. The wide range of PET parameters with different thresholds in this study that showed statistically significant association with DFS corroborates the positive results from previous studies. Among studies on colorectum, the studies by Ogawa et al. and Nakajo et al. investigated multiple PET parameters including SUVmax, MTV, and TLG [26,30]. However, the sample size of the study by Nakajo et al. was relatively small even after including stage IV patients [26]. Thus, the study by Ogawa et al. that included 325 stage I–III colorectal cancer patients primarily treated with surgery is more comparable with this study [30]. They reported MTV and TLG as the PET parameters significantly related to survival in univariate analysis and TLG (cut-off: 341.89) alone as the significant PET parameter in multivariate analysis. Their study is most similar to this study in terms of methodology and results. They reported the same significant PET parameter, TLR, as this study, with modest difference in cut-off levels. However, Ogawa et al. also included stage I patients, while this study included stage II/III only. We deemed that the prognostic value of PET/CT was most necessary in these patients with intermediate risk of recurrence rather than stage I or stage IV with evidently low or high risk of recurrence. Additionally, stage II/III is subjected to adjuvant therapy, and the prognostic role of PET/CT would be maximized if it could help in making treatment decisions. The minor difference in cut-off levels of TLR between the results by Ogawa et. al. and this study may have been due to the narrower inclusion criteria of this study and also the different outcome measures (OS in their study and DFS in this study) as a consequence of difference in follow-up period. We considered that DFS was the endpoint more specifically representative of oncologic outcome because OS may be affected by multiple other non-cancerous factors. Another study reporting TLG as a potential biomarker in colorectal cancer was the one by Lim et al. They included only the metastatic colon cancer patients treated with regorafenib and tested the significance of PET parameters for response evaluation [42]. Although the threshold levels for TLG were different between their study (TLG40%) and this study (TLG2.5), in addition to different clinical setting, it is still notable that TLG was the significant PET parameter in colorectal cancer patients. In the study by Huang et al., TLR was the PET parameter prognostic of survival outcome in stage IIA colorectal cancer [32]. TLR is also a parameter representing both the metabolic activity and metabolic volume, similar to TLG.

The accumulation of ^18^F-FDG in tumors is indicated by SUV. The maximum value of SUV within the tumor is SUVmax, and it is the most commonly documented PET parameter in imaging reports. Although SUVmax is easily reproducible and convenient to use for estimation of the biologic aggressiveness of a tumor, it is a value obtained from a single voxel alone, and non-viable regions within the tumor cannot be accounted for. On the other hand, MTV is the volume of the region with SUV levels greater than a specific threshold, thus reflecting viable tumor volume. However, TLG, the product of SUVmean and MTV, is consistently reported as being prognostic of tumor outcome in locoregionally advanced, resectable colorectal cancer. It suggests that a parameter incorporating both the metabolic activity as well as the metabolic volume of a tumor is important in this specific clinical context.

The remarkable advancement in treatment of locoregionally advanced colorectal cancer is still in progress. In a planned interim report of the FOxTROT phase III trial presented at ASCO 2020, investigating the efficacy and safety of neoadjuvant chemotherapy in 1053 T3-4, N0-2 colon cancer patients, relapse or persistent disease after two years showed a trend toward improvement at 15.6% for neoadjuvant chemotherapy vs. 19.5% for adjuvant chemotherapy (*p* = 0.007) [43]. In addition, the largest randomized phase III trial utilizing neoadjuvant CAPOX in 1370 colon cancer patients is currently recruiting in China [44], of which the final analysis is still ongoing. Additionally, in rectal cancer, total neoadjuvant therapy, involving administration of concurrent chemoradiotherapy followed by adjuvant chemotherapy plus neoadjuvant chemotherapy before surgery, is a promising strategy with superior rates of pathologic complete response compared with standard therapy [45]. Thus, treatment intensification such as neoadjuvant treatment in colon cancer and total neoadjuvant therapy in rectal cancer, as well as treatment de-intensification such as “watch-and-wait” approach, have both proven their efficacy and are being actively considered to maximize the treatment effects [46]. Both strategies narrow the question down to which patients are suitable for intensified or de-intensified treatment. Better understanding of tumor metabolism with preoperative ^18^F-FDG PET/CT before initiation of treatment may support the multidisciplinary team in reaching more individualized and suitable treatment decisions that are in accordance with tumor biology.

This study was a comprehensive demonstration of the significance of PET parameters for prognostication of tumor outcome specifically in locoregionally advanced, resectable colorectal cancer treated with modern systemic therapy. Another noteworthy finding is that among several tested parameters, TLG, a parameter representing both the metabolic activity and metabolic volume, was most significantly related to outcome. We also attempted to find the optimal PET parameter for prognostication by testing several thresholds per parameter. Although we could not reach a definitive conclusion in terms of optimal threshold per parameter, this is another domain for further investigation of prognostication with PET parameters. Previous testing of various SUV thresholds in the context of colorectal cancer have been performed in smaller cohorts with conflicting results [36,38]. Although the inherent limitations of a retrospective study are unavoidable, this was a single institutional study conducted with homogeneous patients under a uniform standard of care. Despite the absence of a validation of the results of this study with an independent dataset, the multiple PET parameters showing an association with tumor outcome signify the potential prognostic value of ^18^F-FDG PET/CT and renders reliability in the extrapolation of the results to locoregionally advanced, resectable colorectal cancer patients.

## 5. Conclusions

Preoperative ^18^F-FDG PET/CT has a potential prognostic value in stage II/III resectable colorectal cancer in the modern era of advanced cancer therapy. Among the several feasible PET parameters tested in this study, TLG, a parameter indicating both metabolic activity and metabolic volume, was the strongest predictor independently associated with DFS.

## Figures and Tables

**Figure 1 cancers-14-00582-f001:**
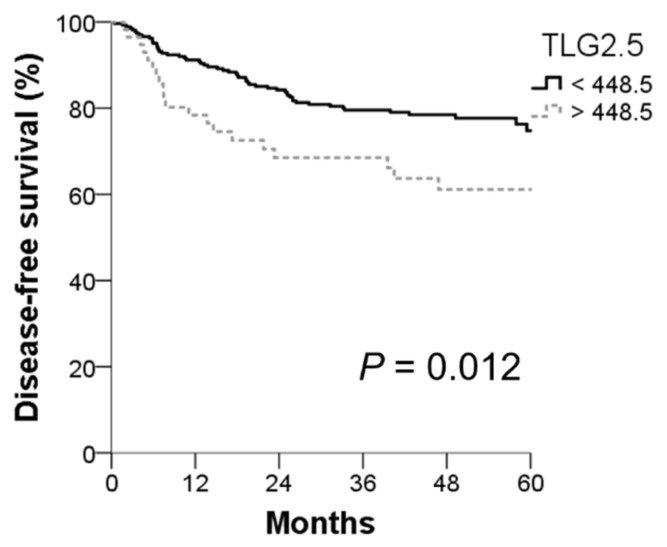
Disease-free survival according to total lesion glycolysis (TLG).

**Table 1 cancers-14-00582-t001:** Baseline characteristics and treatment (*n* = 327).

Clinico-Pathologic Characteristics	*n* (%)
**Age** (median 64, range: 32–86)	
<64	168 (51.4)
≥64	159 (48.6)
**Sex**	
Male	193 (59)
Female	134 (41)
**Sidedness**	
Right	73 (22.5)
Left	140 (43.2)
Rectum	111 (34.3)
**Size**	
≤5 cm	198 (60.6)
>5 cm	129 (39.4)
**Obstruction**	
Yes	34 (10.4)
No	293 (89.6)
**Removed lymph nodes**	
<13	55 (16.8)
≥13	272 (83.2)
**Positive lymph nodes**	
0	168 (51.4)
1–3	92 (28.1)
≥4	67 (20.5)
**Stage**	
II	167 (51.1)
III	160 (48.9)
**CEA** (median 3.24, range: 0–1927 ng/mL)	
<3.24 ng/mL	164 (50.2)
≥3.24 ng/mL	163 (49.8)
**Differentiation**	
Well	10 (3.1)
Moderate	306 (93.6)
Poor	11 (3.4)
**Lymphatic** invasion	
Yes	103 (31.5)
No	223 (68.2)
**Vascular invasion**	
Yes	54 (16.5)
No	273 (83.5)
**Neural invasion**	
Yes	120 (36.7)
No	207 (63.3)
**Margin**	
Positive	18 (5.5)
Negative	308 (94.5)
**KRAS (*n* = 149)**	
Mutated type	62 (41.6)
Wild type	87 (58.4)
**BRAF (*n* = 146)**	
Negative	133 (91.1)
Positive	13 (8.9)
**Microsatellite instability (*n* = 276)**	
Low/Deficient	255 (92.4)
High	21 (7.6)
**Radiotherapy**	
Yes	54 (16.5)
No	273 (83.5)
**Chemotherapy**	
Yes	250 (76.5)
No	77 (23.5)

**Table 2 cancers-14-00582-t002:** Disease-free survival (DFS) rates and hazard ratios for DFS according to cut-off levels for PET parameters.

	Disease-Free Survival					
	Kaplan–Meier Rates				Cox Regression	
		(%)		*p*	Hazard Ratio	*p*
	1 Year	3 Year	5 Year		(95% CI)	
**SUVmax**				0.039		0.041
≤16.2	90.5	79.1	76.8		1.00 (reference)	
>16.2	85.2	74.1	62		1.61 (1.02–2.53)	
**MTV2.5 (cm^3^)**				0.022		0.025
≤93.5	90.5	79.2	74.1		1.00 (reference)	
>93.5	78.1	67.1	60.2		1.91 (1.09–3.36)	
**MTV3 (cm^3^)**				0.011		0.013
≤85.8	98.2	79.1	74.1		1.00 (reference)	
>85.8	77.8	64.9	56.5		2.08 (1.16–3.71)	
**MTV30% (cm^3^)**				0.078		0.087
≤8.5	100	91.2	78.6		1.00 (reference)	
>8.5	87.5	75.8	71.7		2.21 (0.89–5.47)	
**MTV40% (cm^3^)**				0.054		0.060
≤6.4	97.8	90.3	78.4		1.00 (reference)	
>6.4	87.5	75.5	71.3		2.22 (0.97–5.11)	
**TLG2.5**				0.012		0.014
≤448.5	98.1	79.6	74.8		1.00 (reference)	
>448.5	78.4	68.5	61.1		1.90 (1.14–3.15)	
**TLG3**				0.018		0.020
≤647.5	90.4	79.3	73.9		1.00 (reference)	
>647.5	76.7	63.5	58.9		2.03 (1.12–3.69)	
**TLG30%**				0.028		0.030
≤347.6	91.2	79.2	74.4		1.00 (reference)	
>347.6	78.4	70.4	62.5		1.78 (1.06–2.98)	
**TLG40%**				0.056		0.059
≤290.6	90.8	78.9	74.2		1.00 (reference)	
>290.6	79.8	71.6	63.4		1.66 (0.98–2.82)	

**Table 3 cancers-14-00582-t003:** Factors associated with disease-free survival on the Cox proportional hazards model.

	Disease-Free Survival			
	Univariate		Multivariate	
	Hazard Ratio (95% CI)	*p*	Hazard Ratio (95% CI)	*p*
**Age**		0.031		0.041
<64	1.00 (reference)		1.00 (reference)	
≥64	1.64 (1.05–2.57)		1.62 (1.02–2.58)	
**Sex**		0.029		0.017
Male	1.00 (reference)		1.00 (reference)	
Female	0.58 (0.36–0.95)		0.55 (0.33–0.90)	
**Sidedness**		0.283		
Left (including rectum)	1.00 (reference)			
Right	1.43 (0.74–2.76)			
**Size**		0.619		
≤5 cm	1.00 (reference)			
>5 cm	1.12 (0.71–1.77)			
**Stent insertion**		0.723		
No	1.00 (reference)			
Yes	1.14 (0.55–2.37)			
**Removed lymph nodes**		0.076		
<13	1.00 (reference)			
≥13	0.62 (0.36–1.05)			
**Positive lymph nodes**		0.257		
0	1.00 (reference)			
1–3	1.37 (0.81–2.30)			
≥4	1.54 (0.89–2.69)			
**Stage**		0.011		0.097
II	1.00 (reference)		1.00 (reference)	
III	2.31 (1.21–4.44)		1.47 (0.93–2.33)	
**Carcinoembryonic antigen**		0.001		0.001
<3.24 ng/mL	1.00 (reference)		1.00 (reference)	
≥3.24 ng/mL	2.18 (1.34–3.48)		2.19 (1.37–3.51)	
**Differentiation**		0.217		
Well/Moderate	1.00 (reference)			
Poor	1.89 (0.69–5.17)			
**Lymphatic invasion**		0.793		
No	1.00 (reference)			
Yes	0.84 (0.51–1.38)			
**Vascular invasion**		0.861		
No	1.00 (reference)			
Yes	0.95 (0.50–1.79)			
**Neural invasion**		0.740		
No	1.00 (reference)			
Yes	1.08 (0.68–1.71)			
**Margin**		0.064		
Negative	1.00 (reference)			
Positive	2.00 (0.96–4.17)			
**KRAS**		0.238		
Wild type	1.00 (reference)			
Mutated type	1.48 (0.77–2.85)			
**Microsatellite instability**		0.098		
Low/Deficient	1.00 (reference)			
High	0.19 (0.03–1.36)			
**BRAF**		0.766		
Negative	1.00 (reference)			
Positive	1.20 (0.37–3.93)			
**SUVmax**		0.041		0.144
≤16.2	1.00 (reference)		1.00 (reference)	
>16.2	1.61 (1.02–2.53)		1.46 (0.88–2.43)	
**MTV2.5 (cm^3^)**		0.025		0.553
≤93.5	1.00 (reference)		1.00 (reference)	
>93.5	1.91 (1.09–3.36)		0.49 (0.05–5.12)	
**MTV3 (cm^3^)**		0.013		0.579
≤85.8	1.00 (reference)		1.00 (reference)	
>85.8	2.08 (1.16–3.71)		1.97 (0.18–21.74)	
**TLG2.5**		0.014		0.009
≤448.5	1.00 (reference)		1.00 (reference)	
>448.5	1.90 (1.14–3.15)		1.98 (1.19–3.31)	
**TLG3**		0.020		0.949
≤647.5	1.00 (reference)		1.00 (reference)	
>647.5	2.03 (1.12–3.69)		0.96 (0.29–3.19)	
**TLG30%**		0.030		0.896
≤347.6	1.00 (reference)		1.00 (reference)	
>347.6	1.78 (1.06–2.98)		0.90 (0.18–4.57)	

**Table 4 cancers-14-00582-t004:** Previous studies on metabolic parameters from the pretreatment ^18^F-FDG PET/CT of non-metastatic resectable colorectal cancer.

Authors	Year	Site	*n*	Stage	Investigated PET Parameters	Significant PET Parameters	Cut-Off	Method	DFS Rate (%)	*p*	OS Rate (%)	*p*	HR for DFS	*p*	HR for OS	*p*
Lee et al. [25]	2012	Colon/rectum	163	I–IV (IV: *n* = 9)	SUVmax	-	8.6	Median	58 vs. 88.8 (2 yr)	0.53			0.8	0.76		
Byun et al. [29]	2014	Colon/rectum	78	III	SUVmax SUVn (SUVmax of regional nodes)	SUVn	1.2	ROC	58 vs. 86 (5 yr)	0.006			2.97 *	0.026 *		
Ogawa et al. [30]	2015	Colon/rectum	325	I–III	SUVmax SUVmean MTV TLG	MTV TLG	25.23 341.89	ROC			83.8 vs. 91.7 70.1 vs. 92.1 (5 yr)	0.006 0.001			2.42 3.41 *	0.009 0.016 *
Shi et al. [31]	2015	Colon/rectum	107	0–IV (IV: *n* = 5)	SUVmax	SUVmax	11.85	ROC			Median (mo): 37 mo vs. not reached	<0.001				
Huang et al. [32]	2017	Colon/rectum	118	IIA	SUVmax TLR	TLR	6.2	ROC	60.6 vs. 96.9 (5 yr)	<0.001	98.3 vs. 74.3 (5 yr)	0.001	13.365 *	<0.001 *	10.896 *	0.023 *
Nakajo et al. [26]	2017	Colon/rectum	38	I–IV (IV: *n* = 4)	SUVmax SUVmean MTV2.5 TLG	-	12.5 6 15.1 106.2	ROC	87.5 vs. 62.5 81.3 vs. 68 % 68.8 vs. 81.3 75 vs. 75 (5 yr)	0.13 0.48 0.36 0.87			0.89 0.78 1.94 1.14	0.21 0.17 0.37 0.86		
Lee et al. [33]	2018	Colon/rectum	226	I–IV (IV: *n* = 12)	SUV of bone marrow (BM SUV)	BM SUV	1.9	MCS	78.6 vs. 92.1 (2 yr)	0.013			2.94	0.009 *		
Chen et al. [34]	2018	Colon/rectum	90	I–III	SUVmax SUVn	SUVn	1.15	ROC					10.107 *	<0.0001 *		
Martoni et al. [27]	2011	Rectum	80	II–III	SUVmax	-	27	ROC					1.935	0.4		
Lee et al. [35]	2013	Rectum	81	II–III	SUVmax SUVmean MTV TLG	MTV TLG	12 85	Mean					1.416 3.663	0.044 0.012	2.815 20.035 *	0.127 0.017 *
Jo et al. [36]	2014	Rectum	73	I–IV (IV: *n* = 11)	SUVmax MTV2.5 MTV3 MTV3.5 TLG2.5 TLG3 TLG3.5	MTV2.5 MTV3 MTV3.5 TLG2.5 TLG3 TLG3.5	24.1 19.1 15.2 225.3 169.1 143.1	Median	NR NR - NR NR -	0.03 0.04 - 0.02 0.03 -	NR NR NR NR NR NR	0.02 0.03 0.02 0.01 0.02 0.02				
Kim et al. [37]	2015	Rectum	64	II–III	SUVmax SUVmean MTV TLG	MTV	34.9	ROC					57.96 *	0.0078 *	19.70 *	0.0284 *
Bang et al. [38]	2016	Rectum	74	II–III	SUVmax SUVpeak SUVmean MTV TLG	MTV TLG	40% SUVmax threshold (most significant)	30–70% SUVmax threshold, SUVmean of liver + 2SDs and + 3SDs					1.020 1.002	0.003 0.023		
Deantonio et al. [23]	2018	Rectum	100	II–III	SUVmaxSUVmean MTV TLG	-	20.6 11.9 14.9 175	Median	NR NR NR NR (4 yr)	0.65 0.65 0.46 0.26	NR NR NR NR (4 yr)	0.42 0.42 0.24 0.17				
Lovinfosse et al. [39]	2018	Rectum	86	III	SUVmax SUVmean MTV TLG	TLG	550	ROC							6.44 (DSS)	0.0073
Okuno et al. [28]	2018	Rectum	79	II–III	SUVmax TLG	-	14.2 385.6	Mean	78.3 vs. 73.4 67.3 vs. 83.4 (5 yr)	0.95 0.48	100 vs. 89.8 93.3 vs. 96.7 (5 yr)	0.10 0.83				
Alcin et al. [40]	2020	Rectum	115	I–IV (IV: *n* = 23)	SUVmax SUVn	SUVn	3.55	ROC	NR	0.049	NR	0.045				
Choi et al. [41]	2021	Rectum	149	I–III	SUVmax			ROC	Mean (mo):		Mean (mo):					
					MTV	MTV	23.9		51.7 vs. 60	0.005	57.9 vs. 63	0.003	2.47 *	0.042 *	5.65 *	0.028 *
					TLG	TLG	125.84		50 vs. 62.1	0.002	57.1 vs. 64.1	0.044	3.21 *	0.015 *	NR	0.174 *
This study	2021	Colon/rectum	327	II–III	SUVmax MTV2.5 MTV3 MTV30% MTV40% TLG2.5 TLG3 TLG30% TLG40%	SUVmax MTV2.5 MTV3 TLG2.5 TLG3 TLG30%	16.2 93.5 85.8 448.5 647.5 347.6	MCS	62 vs. 76.8 60.2 vs. 74.1 56.5 vs. 74.1 61.1 vs. 74.8 58.9 vs. 73.9 63.4 vs. 74.2 (5 yr)	0.039 0.022 0.011 0.012 0.018 0.028			1.61 1.91 2.08 1.98 * 2.03 1.78	0.041 0.025 0.013 0.009 * 0.020 0.030		

* Multivariate. Abbreviations: FDG-PET, fluoro-D-glucose positron emission tomography; CT, computed tomography; DFS, disease-free survival; OS, overall survival; HR, hazard ratio; SUV, standard uptake value; SUVmax, maximal standard uptake value; MTV, metabolic tumor volume; TLG, total lesion glycolysis; SUVn, SUVmax of regional nodes; ROC, receiver operating characteristic curve; MCS, maximal chi-square analysis; DSS, disease-specific survival; NR, not reported.

## Data Availability

The data presented in this study are available on request from the corresponding author.

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
