# Peer review of "Prognostic Impact of Total Lesion Glycolysis (TLG) from Preoperative 18F-FDG PET/CT in Stage II/III Colorectal Adenocarcinoma: Extending the Value of PET/CT for Resectable Disease"

_cancers, 2022, doi:10.3390/cancers14030582_

Round 1

Reviewer 1 Report

Dear Authors,

Thank you for your submission.

Comments:

  1. In the discussion section, please explain the differences and new sound of this study compared to the study of Ogawa et al. [Ogawa S, Itabashi M, Kondo C, Momose M, Sakai S, Kameoka S. Prognostic Value of Total Lesion Glycolysis Measured by 18F-FDG-PET/CT in Patients with Colorectal Cancer. Anticancer Res. 2015 Jun;35(6):3495-500. PMID: 26026116.].
  2. In line 168, "MSI-low or deficient state", please delete "or deficient" and change it to "MSI-low". because the deficient state is related to MSI-H.

Author Response

Comments:

  1. In the discussion section, please explain the differences and new sound of this study compared to the study of Ogawa et al. [Ogawa S, Itabashi M, Kondo C, Momose M, Sakai S, Kameoka S. Prognostic Value of Total Lesion Glycolysis Measured by 18F-FDG-PET/CT in Patients with Colorectal Cancer. Anticancer Res. 2015 Jun;35(6):3495-500. PMID: 26026116.].

Response:

The authors deeply appreciate your keen comment. This study truly is very similar to the study by Ogawa et al. in terms of methodology and results. However, there are a number of difference between the study by Ogawa et al. vs. this study: 1) inclusion criteria (stage I-III vs. stage II/III); 2) endpoint (OS vs. DFS); and 3) number of threshold per parameter (single vs. multiple).

1) Inclusion criteria (stage I-III vs. stage II/III)

Ogawa et al. included stage I patients as well while this study included only stage II/III colorectal cancer. As mentioned in Introduction, the aim of this study was to assess the prognostic value of preoperative PET/CT in stage II/III colorectal cancer with various reported recurrence rates ranging from 20 to 40%. We deemed that the prognostic value of PET/CT had to be most needfully assessed in these patients with intermediate risk of recurrence (stage II/III) rather than stage I or stage IV with evidently low or high risk of recurrence. In addition, we focused especially on stage II/III because these patients are subjected to adjuvant treatment in the era where treatment intensification strategies such as total neoadjuvant therapy are being actively investigated. The prognostic value of PET/CT would be maximized if PET parameters could aid in treatment decisions.

2) Endpoint (OS vs. DFS)

Ogawa et al. derived the cut-off levels for PET parameters using OS, in contrast with DFS of our study. We considered that DFS was the endpoint more specifically representative of oncologic outcome because OS may be affected by multiple other non-cancerous factors.

3) Number of threshold per parameter (single vs. multiple)

Ogawa et al. used single threshold, 30% of SUVmax, for MTV and TLG, while we tested multiple thresholds including fixed SUV thresholds of 2.5, 3, and relative thresholds of 30% and 40% of SUVmax. We considered the rigorous assessment of PET parameters with multiple thresholds was an addition of new sounds to previous results obtained by Ogawa et al.

However, we strongly agree that your comment is totally valid and have added the above explanation into the manuscript by rephrasing the relevant section in Discussion as follows.

Before:

“Thus the study by Ogawa et al. which included 325 stage I – III colorectal cancer patients primarily treated with surgery is more comparable to this study [30]. They reported MTV and TLG as the PET parameters significantly related to survival in univariate analysis and TLG (cut-off: 341.89) alone as the significant PET parameter in multivariate analysis. With exception of broader inclusion criteria with stage I patients, their study is most similar to this study in terms of patient cohort and results. They also reported the same significant PET parameter, TLR, as this study except for modest difference in cut-off levels. This difference may have been due to the narrower inclusion criteria of this study and also the different outcome measures (OS in their study and DFS in this study) as a consequence of difference in follow-up period.” (line 321-331 in the initial manuscript)

After: [Lines 369-386 in the revised manuscript]

“Thus the study by Ogawa et al. which included 325 stage I – III colorectal cancer patients primarily treated with surgery is more comparable to this study [30]. They reported MTV and TLG as the PET parameters significantly related to survival in univariate analysis and TLG (cut-off: 341.89) alone as the significant PET parameter in multivariate analysis. Their study is most similar to this study in terms of methodology and results. They reported the same significant PET parameter, TLR, as this study with modest difference in cut-off levels. However, Ogawa et al. also included stage I patients while this study included stage II/III only. We deemed that the prognostic value of PET/CT was most necessary in these patients with intermediate risk of recurrence rather than stage I or stage IV with evidently low or high risk of recurrence. Also, stage II/III are subjected to adjuvant therapy and the prognostic role of PET/CT would be maximized if it could help making treatment decisions. The minor difference in cut-off levels of TLR between the results by Ogawa et. al. and this study may have been due to the narrower inclusion criteria of this study and also the different outcome measures (OS in their study and DFS in this study) as a consequence of difference in follow-up period. We considered that DFS was the endpoint more specifically representative of oncologic outcome because OS may be affected by multiple other non-cancerous factors.” (lines 369-386 in the revised manuscript)

Thank you again for your comment which has greatly improved the manuscript by clearing up ambiguities regarding previous reports.

  1. In line 168, "MSI-low or deficient state", please delete "or deficient" and change it to "MSI-low". because the deficient state is related to MSI-H.

Response:

Thank you so much for amending such an elementary error. The misnomer has been corrected.

The authors once again are grateful for your sharp observation.

Reviewer 2 Report

Review:cancers-1545798

It is a nice study showing good evidence of FDG-PET/CT as a prognostic indicator for colorectal adenocarcinoma. As a manuscript, it has included several unclear points and appropriate revision is recommended.

  1. Abstract and conclusion: In Table 3 and Figure 1, It is clearly showed that TLG2.5 was a significant predictor for prognosis, but TLG3 and TLG30% did not. While the author generalized that TLG was a strong independent predictor. I think there are no reason to generalize the results of TLG2.5 to all TLG. I think the author should be more strict to the results and conclusion.
  2. Patients. They had 486 patients at the start line. While, analysis was performed on 327 patients. 159 patients (32.5%) were missing. With which reason were these patients excluded from the analysis? It may be a sufficient number to modify or even to decorate the results of the analysis. In order to have reliability for your analysis, you should present the detail of these missing or excluded patients.
  3. Table 2. This table seems to be presented duplicated.
  4. Results section. There were almost complete duplications to the contents of the tables. I would like to recommend more concise description on the emphasizing points.
  5. Table 3. Sidedness; Left and Right. Did you excluded rectal cancer from this analysis or included in the Left? Please clarify.
  6. Table 3. The best prognostic indicator was TLG2.5, its p value was 0.009. It looks like not better than the CEA. Is it true? Is it a limitation?
  7. Discussion. Line 282. (Table 4). I cannot find Table 4 in your manuscript. Line370. Please correct grammatical error.

Author Response

Comments:

It is a nice study showing good evidence of FDG-PET/CT as a prognostic indicator for colorectal adenocarcinoma. As a manuscript, it has included several unclear points and appropriate revision is recommended.

Response:

The authors deeply appreciate your thorough review and thoughtful comments which have substantially enhanced the quality of the manuscript.

Abstract and conclusion: In Table 3 and Figure 1, It is clearly showed that TLG2.5 was a significant predictor for prognosis, but TLG3 and TLG30% did not. While the author generalized that TLG was a strong independent predictor. I think there are no reason to generalize the results of TLG2.5 to all TLG. I think the author should be more strict to the results and conclusion.

Response:

Thank you for your perceptive comment. Just as you have pointed out, we regarded TLG2.5 itself as a predictor and have not thought of grouping it with other TLG with different threshold levels such as TLG3 or TLG30%. We initially considered the results with TLG2.5 were valid enough to draw conclusions that TLG is the most significant parameter among different PET parameters such as SUVmax and MTV in stage II/III colorectal cancer. We thought that which thresholds within TLG best demonstrates the prognosis was another area of research outside the scope of this study. However, thanks to your insightful comment, we have included your concerns in the relevant section of Discussion as follows.

Before:

“When the PET parameters significantly related to DFS in univariate analysis were tested for independent association with DFS in multivariate analysis including clinicopathologic factors, TLG2.5 was the only significantly associated PET parameter (P = 0.009). SUVmax is a parameter for metabolic activity and MTV is a volumetric parameter while TLG2.5 is a parameter indicating both the metabolic activity and metabolic volume; which may explain the maintenance of statistical significance only for TLG2.5 (cut-off: 448.5) at multivariate analysis. For locoregionally advanced, resectable colorectal cancer, the PET parameter demonstrating both the metabolic activity and metabolic volume such as TLG significantly prognosticated DFS.” (line 305 -313 in the initial manuscript)

After:

“When the PET parameters significantly related to DFS in univariate analysis were tested for independent association with DFS in multivariate analysis including clinicopathologic factors, TLG2.5 was the only significantly associated PET parameter (P = 0.009). SUVmax is a parameter for metabolic activity and MTV is a volumetric parameter while TLG2.5 is a parameter indicating both the metabolic activity and metabolic volume; which may explain the maintenance of statistical significance only for TLG2.5 (cut-off: 448.5) at multivariate analysis. Although TLG2.5, TLG3, and TLG 30% were significantly related to DFS in univariate analysis and TLG2.5 was the only TLG which retained significance in multivariate analysis, which threshold within TLG best demonstrates the prognosis remains to be seen in future studies. For locoregionally advanced, resectable colorectal cancer, the PET parameter demonstrating both the metabolic activity and metabolic volume such as TLG significantly prognosticated DFS.” (line 343-361 in the revised manuscript)

We once again thank you for improving the manuscript with your robust logical reasoning.

Patients. They had 486 patients at the start line. While, analysis was performed on 327 patients. 159 patients (32.5%) were missing. With which reason were these patients excluded from the analysis? It may be a sufficient number to modify or even to decorate the results of the analysis. In order to have reliability for your analysis, you should present the detail of these missing or excluded patients.

Response:

The authors apologize for the misleading language. What we meant was that among 486 colorectal patients who underwent primary surgical resection from 2009 to 2013, 327 patients satisfied the inclusion criteria and were analyzed in this study. Thanks to your comment, we have rephrased the Results section 3.1 as follows.

Before:

“A total of 327 patients diagnosed with resectable stage II – III colorectal cancer who underwent preoperative FDG PET/CT were analyzed (Table 1).” (line 194-195 in the initial manuscript)

After:

“Among 486 colorectal patients who underwent primary surgical resection, 327 patients diagnosed with resectable stage II/III colorectal cancer with preoperative FDG PET/CT satisfied the inclusion criteria and were analyzed in this study (Table 1).”(line 194-196 in the revised manuscript)

Thank you again for clarifying vague dictions.

Table 2. This table seems to be presented duplicated.

Response:

Thank you so much for your perseverance which has discovered the human error that we have inadvertently missed. The duplicated table has been deleted.

Results section. There were almost complete duplications to the contents of the tables. I would like to recommend more concise description on the emphasizing points.

Response:

Thank you for your comment which has greatly reduced redundancy of the manuscript. The Results sections has been rewritten as below.

Before:

3.1 Patient, tumor, and treatment characteristics

“Median age was 64 years (range: 32 – 86). Male were more predominant (64%). The sidedness of disease in order of frequency was as follows: left (43.2%), rectum (34.3%), and right (22.5%), respectively. Approximately 60% of tumors were sized £ 5 cm and 10% presented with obstruction. Majority of patients (83.2%) had ³ 13 lymph nodes removed at surgery. Median preoperative CEA level was 3.24 ng/mL (range: 0 – 1,927). Histology was moderately differentiated in 93.6%. Lymphatic, vascular, and neural invasion was positive in 31.5%, 16.5%, and 36.7%, respectively. Clear resection margin was achieved in 94.5%. Among 149 patients tested for KRAS mutation, 41.6% (62/149) were mutated. Only 8.9% (13/146) were BRAF positive among 146 tested patients. MSI status was tested in 276 patients and 7.6% (21/276) were MSI-high. Approximately half (N = 54) of rectal cancer patients (N = 111) underwent radiotherapy and over three quarter of patients (76.5%) received chemotherapy.” (line 194-206 in the initial manuscript)

3.4 Factors associated with DFS

“The PET parameters significantly associated with DFS in univariate analyses; SUVmax, MTV2.5, MTV3, TLG2.5, TLG3, and TLG30% were analyzed with other clinicopathologic factors for association with DFS (Table 3). The factors significantly related to DFS in univariate Cox regression analysis were age, sex, stage, preoperative CEA level, SUVmax, MTV2.5, MTV3, TLG2.5, TLG3, and TLG30% (Table 3). Older patients were more likely to recur than younger patients (P = .031). Female patients were less likely to experience recurrence than male (P = .029). Stage III was more than twice as likely to recur compared to stage II (P = .011). The results were similar for patients with higher preoperative CEA level (HR 2.18) in comparison with patients with lower CEA level (P = .001). Patients with higher SUVmax of > 16.2 had over 60% greater risk of recurrence compared to patients with lower SUVmax of £ 16.2 (P = .041). Patients with MTV2.5 > 93.5 cm3 had nearly twice the risk of recurrence than the lower counterpart (P = .025). Patients with MTV3 > 85.8 cm3 had over twice the risk of relapse compared to the lower counterpart. The results were similar in patients with higher TLG2.5 of > 448.5 (HR 1.90; P = .014) and TLG3 > 647.5 (HR 2.03; P = .020). Patients with higher TLG30% of > 347.6 also had over three quarter the risk of recurrence than the lower counterpart (HR 1.78; P = .030). Patients with positive margin (P = .064) and number of removed lymph nodes less than 13 (P = .076) had higher risk of recurrence but the difference was marginal.” (line 229-247 in the initial manuscript)

“Among the factors significantly associated with DFS in univariate analysis, age, sex, preoperative CEA level, and TLG2.5 sustained statistically significant difference in multivariate analysis (Table 2). Patients with age ³ 64 had a HR of 1.62 for recurrence compared to younger patients (P = .041). Female patients had 55% of the risk for recurrence compared to male (P = .017). The results for preoperative CEA level was similar with univariate analysis and patients with CEA level ³ 3.24 ng/mL had more than twice the risk of recurrence (HR 2.19) than CEA < 3.24 ng/mL (P = .001). Patients with TLG2.5 > 448.5 were nearly twice as likely to recur (HR 1.98, 95% CI 1.19 – 3.31) compared to patients with TLG2.5 £ 448.5 with strong statistical significance (P = .009). The 1-, 3-, and 5-year DFS rates for patients with TLG2.5 £ 448.5 were 98.1%, 79.6%, and 74.8%, significantly higher than 78.4%, 68.5%, and 61.1% of patients with TLG2.5 > 448.5, respectively (Figure 1).”

(line 250-260 in the initial manuscript)

After:

3.1 Patient, tumor, and treatment characteristics

“Male were more predominant and majority of the diseases were left-sided. Although over half of the tumors were small (£ 5 cm), 10% presented with obstruction. Majority of patients had ³ 13 lymph nodes removed at surgery. Most of the tumors were moderately differentiated. Clear resection margin was achieved in majority of patients. While majority of the patients were tested for MSI, less than half of the patients were tested for KRAS or BRAF mutation. Approximately half of rectal cancer patients underwent radiotherapy and over three quarter of patients received chemotherapy.” (line 198-205 in the revised manuscript)

3.4 Factors associated with DFS

“The PET parameters significantly associated with DFS in univariate analyses; SUVmax, MTV2.5, MTV3, TLG2.5, TLG3, and TLG30% were analyzed with other clinicopathologic factors for association with DFS (Table 3). The factors significantly related to DFS in univariate Cox regression analysis were age, sex, stage, preoperative CEA level, SUVmax, MTV2.5, MTV3, TLG2.5, TLG3, and TLG30% (Table 3). Older patients were more likely to recur than younger patients. Female patients were less likely to experience recurrence than male. Stage III was more than twice as likely to recur compared to stage II. The results were similar for patients with higher preoperative CEA level in comparison with patients with lower CEA level. Patients with higher SUVmax had over 60% greater risk of recurrence compared to patients with lower SUVmax. Patients with higher MTV2.5 had nearly twice the risk of recurrence than the lower counterpart. Patients with higher MTV3 had over twice the risk of relapse compared to the lower counterpart. The results were similar in patients with higher TLG2.5 and TLG3. Patients with higher TLG30% also had over three quarter the risk of recurrence than the lower counterpart. Patients with positive margin and number of removed lymph nodes less than 13 had higher risk of recurrence but the difference was marginal.”(line 239-255 in the revised manuscript)

“Among the factors significantly associated with DFS in univariate analysis, age, sex, preoperative CEA level, and TLG2.5 sustained statistically significant difference in multivariate analysis (Table 2). Older patients had over 60% greater risk of recurrence than younger patients. Female patients had 55% of the risk for recurrence compared to male. The results for preoperative CEA level was similar with univariate analysis and patients with higher CEA level had more than twice the risk of recurrence than the lower counterpart. Patients with higher TLG2.5 were nearly twice as likely to recur compared to patients with lower TLG2.5 with strong statistical significance. The 1-, 3-, and 5-year DFS rates for patients with lower TLG2.5 were 98.1%, 79.6%, and 74.8%, significantly higher than 78.4%, 68.5%, and 61.1% of patients with higher TLG2.5, respectively (Figure 1).”(line 271-280 in the revised manuscript)

Thank you again for greatly clarifying the Results section.

Table 3. Sidedness; Left and Right. Did you excluded rectal cancer from this analysis or included in the Left? Please clarify.

Response:

Thank you for your precise comment. Rectal cancer was included in the left-sided cancer for binary analysis. Table 3 has been revised relevantly as follows (please refer to the revised Table 3).

Before: “Left”        à            After: “Left (including rectum)”

Table 3. The best prognostic indicator was TLG2.5, its p value was 0.009. It looks like not better than the CEA. Is it true? Is it a limitation?

Response:

The authors apologize for confusing expression. What we meant was that TLG2.5 was the best prognostic indicator among other PET parameters. As you have keenly observed, CEA had a P-value of .001 and of course, it is the strongest prognostic factor. However, because CEA is an already well-known prognostic factor, we wanted to emphasize the additional value of TLG2.5 in prognostication of stage II/III colorectal cancer. We have revised the Discussion according to your comments as follows.

Before:

“When the PET parameters significantly related to DFS in univariate analysis were tested for independent association with DFS in multivariate analysis including clinicopathologic factors, TLG2.5 was the only significantly associated PET parameter (P = 0.009).” (line 305-308 in the initial manuscript)

After:

“When the PET parameters significantly related to DFS in univariate analysis were tested for independent association with DFS in multivariate analysis including other clinicopathologic factors, TLG2.5 was the only significantly associated PET parameter (P = .009). The prognostic factor with most powerful association was CEA level (P = .001), which is an already well-established prognostic factor. This result again demonstrates the validity of our data by corroborating previous observations. Our data sufficiently suggests the additional value of PET parameters for prognostication. ” (line 343-350 in the revised manuscript)

Discussion. Line 282. (Table 4). I cannot find Table 4 in your manuscript.

Response:

Wow, we are shocked by the insensibility of the error. Thank you so much for pointing it out. We have added Table 4 into the manuscript.

Line370. Please correct grammatical error.

Response:

The authors are deeply grateful for your detailed review of manuscript. The grammatical error has been corrected.